# The implementation of a nutrition protocol in a surgical intensive care unit; a randomized controlled trial at a tertiary care hospital

**Pornrat Chinda[1], Pulyamon Poomthong[1], Puriwat Toadithep[2], Chayanan Thanakiattiwibun[2], Onuma Chaiwat[2]***

**1** Division of Critical Care Medicine, Department of Medicine, Faculty of Medicine, Siriraj Hospital, Mahidol University, Bangkok, Thailand, **2** Department of Anesthesiology, Faculty of Medicine, Siriraj Hospital, Mahidol University, Bangkok, Thailand

* onuma.cha@mahidol.ac.th

## Abstract

**Data Availability Statement:** All relevant date are within the paper and its Supporting Information files.

### Background

Malnutrition in critically ill patients is linked with significant mortality and morbidity. However, it remains controversial whether nutrition therapy protocols are effective in improving clinical outcomes. The present study aimed to evaluate the effectiveness of a surgical ICU nutrition protocol, and to compare the hospital mortality, hospital LOS, and ICU LOS of protocol and non-protocol groups.

### Methods

A randomized controlled trial was conducted at the Surgical ICU, Siriraj Hospital. **The** nutrition administration of the control group was at the discretion of the attending physicians, whereas that of the intervention group followed the "Siriraj Surgical ICU Nutrition Protocol". Details of the demographic data, nutritional data, and clinical outcomes were collected.

### Results

In all, 170 patients underwent randomization, with 85 individuals each in the protocol and non-protocol groups. More than 90% of the patients in both groups were at risk of malnutrition, indicated by a score of $\geq 3$ on the Nutritional Risk Screening 2002 tool. **The** average daily calories of the 2 groups were very similar (protocol group, 775.4±342.2 kcal vs. control group, 773.0±391.9 kcal; $p = 0.972$). **However**, the median time to commence enteral nutrition was shorter for the protocol group (1.94 days) than the control group (2.25 days; $p = 0.503$). Enteral nutrition was provided within the first 48 hours to 53.7% of the protocol patients vs. 47.4% of the control patients ($p = 0.589$). In addition, a higher proportion of the protocol patients (36.5%) reached the 60% calorie-target on Day 4 after admission than that for the non-protocol group (25.9%; $p = 0.136$). **All** other clinical outcomes and nutrition-related complications were not significantly different.

**Funding:** The author(s) received no specific funding for this work.

**Competing interests:** The authors have declared that no competing interests exist.

## Conclusions

The implementation of the nutrition protocol did not improve the feeding effectiveness or clinical outcomes as compared to usual nutrition management practices of the Surgical ICU.

## Background

Malnutrition in critically ill patients is associated with heightened mortality and morbidity rates. Up to 40% of adult patients are seriously malnourished at the time of their hospital admission, and two-thirds of all patients experience a deterioration in their nutritional status during their hospital stay.[1] The catabolic response in acute critically ill patients is more pronounced than that due to fasting in healthy individuals; it is superimposed by immobilization coupled with marked inflammatory and endocrine stress responses. Skeletal muscle wasting and weakness are associated with a prolonged need for mechanical ventilation, rehabilitation, and post-intensive care unit (ICU) disability.[2] Therefore, adequate nutrition therapy is important as an integral component in the treatment of critically ill patients. A previous multi-center study at 3 surgical ICUs (SICUs) in Thailand reported ineffective nutritional support in terms of the total calories received and the late initiation of enteral nutrition (EN).[3] In addition, a further study on the same population demonstrated that the combination of EN and parenteral nutrition (PN) demonstrated a protective effect on mortality.[4]

There have been several recommendations on medical nutrition therapy for ICU patients. [5, 6] Recently, the European Society for Parenteral and Enteral Nutrition[7] published revised guidelines for clinical nutrition in ICUs. However, it is necessary to use a local protocol that both follows the recommendations derived from evidence-based medicine and is adapted to local administrative practices. In a study by Barr et al. at the medical-surgical ICU of a university teaching hospital and an affiliated Department of Veterans Affairs Hospital in the USA, an evidence-based nutrition management protocol increased the likelihood that ICU patients would receive EN and shortened their duration of mechanical ventilation.[1] In addition, Heyland and colleagues[8] demonstrated that the Enhanced Protein-Energy Provision via Enteral Route Feeding protocol could increase the amount of calories and protein received. However, other recent studies did not demonstrate positive clinical outcomes for nutrition protocol use in ICUs in terms of the duration of mechanical ventilation, the incidence of nosocomial infections, and the length of stay (LOS).[9, 10] Given that, it is still controversial whether nutrition therapy protocols are effective in improving clinical outcomes. The aims of this study were therefore twofold. The first was to develop a nutrition protocol that was based on both the evidence-based guidelines and our administrative policies. The second goal was to compare the effectiveness of the nutrition therapy and the clinical outcomes including nutritional related complication, hospital and ICU mortality and length of stay of two groups: critically ill patients whose therapy followed the nutrition protocol, and critically ill patients whose therapy depended upon the clinical judgements of their attending physicians.

## Materials and methods

A single-blind randomized controlled trial was conducted at the SICU of Siriraj Hospital November 2015–February 2017. The study was approved by the Institutional Review Board of the Faculty of Medicine, Siriraj Hospital, Mahidol University, Thailand (Si 618/2015). Written

informed consent was obtained from each participant before their entry into the study. The trial was also registered with the Thai Clinical Trials Registry (TCTR20160510003).

## Study population

The study population comprised adult patients aged 18 years or older who were admitted to the SICU, Siriraj Hospital, with an expected stay exceeding 48 hours. Excluded were patients who were tolerating an oral diet or were scheduled to return to an oral intake within 24 hours; were receiving palliative care; were moribund and not expected to survive 6 hours; were brain dead or suspected to be brain dead; had been admitted directly from another ICU; or were foreigners. Data were collected November 2015–February 2017.

## Measurement instruments and data collection

In the case of males, the formula used to calculate their predicted body weight (in kg) was $(50.0 + 0.91^*(\text{height in cm}– 152.4))$. As to females, the predicted body weight (in kg) was given by $(45.5 + 0.91^*(\text{height in cm}– 152.4))$ [11]. The identification of patients at nutritional risk was determined using the scores for the Nutritional Risk Screening 2002 (NRS-2002) tool[12], the Malnutrition Universal Screening Tool (MUST)[13], and the Nutrition Risk in the Critically Ill (NUTRIC)[14] screening tool. A score for the NRS-2002 of $\geq 3$, for MUST of $\geq 2$, or for NUTRIC of $\geq 5$ was defined as a malnourished condition that required medical nutrition therapy. The effectiveness of the therapy was reported in terms of the time to commence it, the percentage of patients who received EN within the first 48 hours, the total daily calories received, and the proportion of patients receiving $\geq 60\%$ of the target calories on Day 4 of their ICU admission. The nutrition-related complications were expressed as the percentage of patients who had hyper/hypoglycemia, vomiting, aspiration, abdominal distention, or a gastric residual volume $\geq 250$ ml. Lastly, the clinical outcomes were the ICU mortality rate, the in-hospital mortality rate, the ICU/hospital LOS, the duration of mechanical ventilation, and the incidence of new infections developed in the ICU.

The primary outcome of this study was the effectiveness of the nutrition therapy provided by the Siriraj SICU Nutrition Protocol compared with the non-nutrition protocol. The secondary outcomes were the hospital mortality rates, hospital LOS, and ICU LOS of the protocol and non-protocol groups. In accordance with Siriraj Hospital guidelines, both groups were administered standard therapy to prevent catheter-associated blood-stream infections as well as a ventilator-associated pneumonia bundle.

## Procedure

Computerized randomization of the patients into the two study groups was done by researcher (PC, PP) in blocks of four and by using sealed envelopes. Nutrition administration in the control (non-protocol) group was done in accord with the discretion of the attending physicians. In the intervention (protocol) group, however, the nutrition management followed the "Siriraj Surgical ICU Nutrition Protocol". The patient was blind to the allocation.

After enrollment, the baseline characteristics and nutritional status data were collected. The NRS-2002 and MUST scores were used to evaluate the baseline nutritional data, while the NUTRIC score was subsequently calculated to determine the need for supplemental parenteral nutrition (SPN). In the case of the protocol group, the 100%-target calories were defined as 30 kcal per kg of predicted body weight. The Siriraj SICU Nutrition Protocol (S1 Fig) was started with EN within 24 hours (with the permission of the surgeons), provided there were no contra-indications (such as active shock with high-dose vasopressor; tolerating an adequate oral diet, or requiring less than 24 hours to begin an oral diet; and receiving palliative care). The

acceptable surgical conditions for which EN was not allowed to commence within 24 hours are detailed in Diagram 1; those conditions were re-assessed with the surgeons every 12 hours. Although total PN might be considered for those conditions, the patients were still re-assessed for EN eligibility every 12 hours. EN was commenced with a full-strength (1:1) polymeric formula. The goal was 80% of the target calories within 72 hours. A prokinetic agent was administered if that goal was not achieved; alternatively, postpyloric feeding might be considered on a case-by-case basis. In the event that the total daily calorie intake was still < 60% of the target on Day 4 of admission, a patient's NUTRIC score would then be redetermined and the SPN Protocol would be observed (S2 Fig). If the NUTRIC score was ≥ 5, partial PN was initiated to reach the 100% calorie-target. On the other hand, if the NUTRIC score was < 5, the EN was continued until Day 8. Partial PN was then commenced if the calories from the EN were still < 60% of the target. The management of gastrointestinal intolerance is illustrated in S3 Fig.

## Statistical analysis

To compare the effectiveness of the nutrition administration protocol in the SICU, we used information drawn from a previous SICU database. It was estimated that 50% of the patients in the control group and 75% in the protocol group would achieve 100% of the target calories with statistical significance.

The sample size calculation was performed with the n4studies software (version 1.4.1) by using the formula for randomized–controlled-trial binary data with a subsequent continuity correction. The sample size in each arm was initially calculated to be 77; following the continuity correction, the size was revised to 85 for each group. The demographic and nutrition variables were presented as mean ± standard deviation and median (interquartile range) for the continuous data, and frequency and percentage for the categorical data. Group comparisons were performed by using the independent Student's t-test, Mann–Whitney U test, chi-squared test, or Fisher's exact test, as appropriate. A two-sided alpha level of 0.05 was required for statistical significance. The data were analyzed by using the Statistical Package for Social Sciences for Windows, version 18 (SPSS Inc., Chicago, IL, USA).

## Results

Of the 182 patients assessed for eligibility, 12 were excluded (ten were in the SICU for less than 48 hours; one withdrew consent; and another returned to oral intake within 24 hours). The 170 remaining patients underwent randomization, with 85 individuals in each study arm (Fig 1). The vast majority of the baseline demographic data of the protocol and non-protocol groups were not statistically different. That data comprised age; sex; underlying diseases; body mass index (BMI); albumin level; type of operation; sepsis at the time of admission; elective or emergency surgery; American Society of Anesthesiologists (ASA) physical status; Acute Physiology, Age, Chronic Health Evaluation (APACHE) II score; Sequential Organ Failure Assessment (SOFA) score; and the NRS-2002, MUST, and NUTRIC scores. However, significantly more protocol patients than non-protocol patients had hypertension (protocol group, 68.2%, vs. control group, 50.6%; $p$ = 0.028) and vascular operations (protocol group, 25.9%, vs. control group, 13.6%; $p$ = 0.048; Table 1).

The severity of malnutrition did not significantly differ between the groups. Almost all patients in both groups were at risk of malnutrition (indicated by an NRS-2002 score ≥ 3), and more than 50% of the patients in each group had a NUTRIC score ≥ 5. Regarding the effectiveness of the nutrition therapy, the average calories received per day by both groups were not significantly different (protocol group, 775.4±342.2 kcal vs. control group, 773.0 ±391.9 kcal; $p$ = 0.972). The median time to commence EN was shorter for the protocol

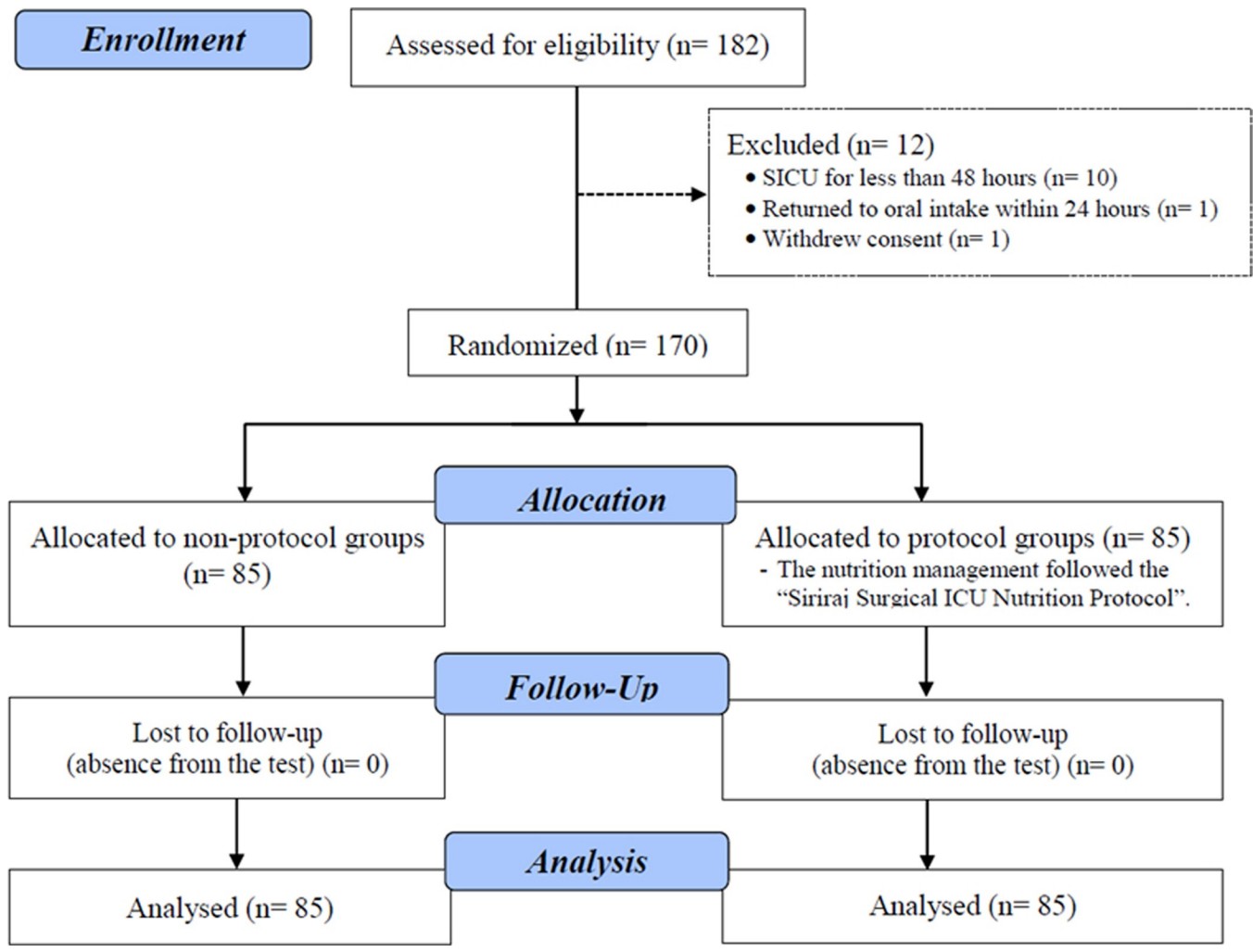

**Fig 1. Consort flowchart.**

patients than the control patients (1.94 and 2.25 days, respectively; $p$ = 0.503]. More than half of the patients in the two groups were given EN combined with PN (protocol group, 64.3% vs. control group, 57.1%; $p$ = 0.430). In addition, a significantly higher proportion of protocol patients than control patients received EN within the first 48 hours (53.7% and 47.4%, respectively; $p$ = 0.589). On the other hand, although a greater proportion of patients in the protocol group (36.5%) than in the control group (25.9%) reached the 60% calorie-target on Day 4 after admission, the difference was not statistically significant ($p$ = 0.136). Nearly half of the patients in both groups were unable to receive EN due to their surgical conditions. Moreover, the main reason for interrupting or stepping up the EN was the patients' surgical conditions including gastroparesis, ileus and surgical complications (leakage and sepsis), followed by hemodynamic instability. The use of prokinetic agents in the groups did not differ (Table 2).

The ICU mortality rates (protocol, 12% vs. control, 18%) and hospital mortality rates (protocol, 27% vs. control, 26%) were not statistically different. The median ICU LOS was 6 days for each group (protocol group, 6.2 (4.3, 13.3) days vs. control group, 6.3 (3.6, 13.1) days). The hospital LOS and duration of mechanical ventilation were also not significantly different (Table 3). The prevalence of nutrition–related complications and the rates of new infections in

**Table 1. Baseline characteristics of the patients.**

| Characteristic | Protocol (n = 85) | Control (n = 85) | p-value |
|---|---|---|---|
| Age (years), (mean ± SD) | 66.6 ± 15.8 | 62.1 ± 16.0 | 0.065 |
| Male, n (%) | 46 (54.1) | 47 (55.3) | 1.000 |
| Body weight (kg), (mean ± SD) | 61.2 ± 17.6 | 59.9 ± 18.2 | 0.637 |
| Body mass index (kg/m$^2$), (mean ± SD) | 23.3 ± 5.8 | 23.6 ± 6.8 | 0.751 |
| Body mass index (kg/m$^2$), n (%) | | | 0.484 |
| < 20 | 22 (25.9) | 27 (31.8) | |
| ≥ 20 to < 25 | 42 (49.4) | 32 (49.4) | |
| ≥ 25 to < 30 | 13 (15.3) | 17 (20.0) | |
| ≥ 30 | 8 (9.4) | 9 (10.6) | |
| DM, n (%) | 34 (40.0) | 24 (28.2) | 0.145 |
| HT, n (%) | 58 (68.2) | 43 (50.6) | 0.028 |
| CKD, n (%) | 24 (28.2) | 16 (18.8) | 0.205 |
| Cirrhosis, n (%) | 8 (9.4) | 10 (11.8) | 0.804 |
| Stroke, n (%) | 19 (22.4) | 11 (12.9) | 0.158 |
| Cancer, n (%) | 25 (29.4) | 32 (37.6) | 0.330 |
| COPD, n (%) | 3 (3.5) | 3 (3.5) | 1.000 |
| Immunocompromised, n (%) | 2 (2.4) | 8 (9.4) | 0.050 |
| Type of operation, n (%) | | | |
| Intra-abdominal | 43 (53.1) | 52 (64.2) | 0.202 |
| Extra-abdominal | 17 (21.0) | 18 (22.2) | 0.849 |
| Vascular | 21 (25.9) | 11 (13.6) | 0.048 |
| ASA class ≥ 3, n (%) | 69 (86.3) | 76 (93.8) | 0.122 |
| APACHE II score, (mean ± SD) | 22.6 ± 7.5 | 22.0 ± 6.8 | 0.600 |
| SOFA score, (mean ± SD) | 8.2 ± 3.7 | 7.9 ± 3.5 | 0.669 |
| Emergency at admission, n (%) | 58 (71.6) | 57 (70.4) | 1.000 |
| Septic shock, n (%) | 33 (38.8) | 29 (34.1) | 0.633 |
| Albumin on admission, (mean ± SD) | 2.9 ± 0.6 | 3.0 ± 0.7 | 0.176 |
| NRS-2002 ≥ 3, n (%) | 80 (94.1) | 83 (97.6) | 0.443 |
| MUST ≥ 2, n (%) | 12 (14.1) | 18 (21.2) | 0.314 |
| NUTRIC ≥ 5, n (%) | 59 (69.4) | 47 (55.3) | 0.081 |

Abbreviations: DM, diabetes mellitus; HT, hypertension; CKD, chronic kidney disease; COPD, chronic obstructive pulmonary disease; APACHE II score, acute physiology and chronic health evaluation score; ASA class, American Society of Anesthesiologists Classification; SOFA score, Sequential Organ Failure Assessment score; NRS-2002, Nutrition Risk Screening 2002; MUST, Malnutrition Universal Screening Tool; NUTRIC, Nutrition Risk in Critically Ill.

the ICU of the 2 groups were also similar. There were no statistically significant differences in the groups' nutrition-related complications (such as hyperglycemia, hypoglycemia, vomiting, and aspiration) (Table 4).

Although the Siriraj SICU Nutrition Protocol was administered to 85 patients, only 32 (37.6%) could receive EN within 24 hours of their admission. According to the protocol, EN was unable to be implemented for the remaining 53 patients (66%) during the initial 24-hour period due to their particular surgical conditions. Ninety percent of those patients (48 out of 53) were therefore given total PN as a substitute in that period.

## Discussion

This randomized controlled trial demonstrated that there was no significant difference between the protocol-fed and non-protocol-fed groups in terms of the effectiveness of feeding.

**Table 2. Nutritional outcomes.**

| Outcome | Protocol (n = 85) | Control (n = 85) | p-value |
|---|---|---|---|
| EN only, n (%) | 13 (15.5) | 9 (10.7) | 0.360 |
| PN only, n (%) | 17 (20.2) | 27 (32.1) | 0.079 |
| EN and PN, n (%) | 54 (64.3) | 48 (57.1) | 0.430 |
| Time until EN feeding (days), median (IQR) | 1.9 (0.8–3.3) | 2.3 (1.2–3.6) | 0.503 |
| Time to start of PN (days), median (IQR) | 0.5 (0.1–1.2) | 0.6 (0.1–1.6) | 0.567 |
| EN within 48 hours, n (%) | 36 (53.7) | 27 (47.4) | 0.589 |
| Calories received (kcal/day), (mean ± SD) | 775.4 ± 342.2 | 773 ± 391.9 | 0.972 |
| Calories received (kcal/kg/day), (mean ± SD) | 13.5 ± 6.3 | 13.5 ± 7.6 | 0.936 |
| 60% calorie target received on Day 4, n (%) | 31 (36.5) | 22 (25.9) | 0.136 |
| Protein received (g/day), (mean ± SD) | 40.3 ± 19.7 | 47.4 ± 22.7 | 0.045 |
| Protein received (g/kg/day), (mean ± SD) | 0.7 ± 0.3 | 0.8 ± 0.4 | 0.039 |
| 3-in-1 formula, n (%) | 33 (45.8) | 37 (49.3) | 0.742 |
| Prokinetic, n (%) | 22 (25.9) | 19 (22.4) | 0.591 |
| NPO reasons, n (%) | | | 0.339 |
| Surgical condition | 40 (47.6) | 46 (54.1) | |
| ICU procedure | 0 | 0 | |
| Unstable hemodynamics | 12 (14.3) | 8 (9.4) | |
| Interrupt step EN, n (%) | | | 0.936 |
| Surgical condition | 24 (35.3) | 19 (32.2) | |
| ICU procedure | 5 (7.4) | 6 (10.2) | |
| Unstable hemodynamics | 4 (5.9) | 3 (5.1) | |
| GI intolerance | 0 | 0 | |

Abbreviations: EN, enteral nutrition; PN, parenteral nutrition; IQR, interquartile range; SD, standard deviation; NPO, nothing per oral; ICU, intensive care unit.

Their efficacies were assessed by the time to commence medical nutrition therapy, the total daily calories received, the proportion of patients receiving EN within the first 48 hours, and the proportion achieving ≥ 60% of the target calorie intake on Day 4 after ICU admission. However, a higher percentage of the protocol patients received the combination of EN and PN. The protocol we created focused on the administration of SPN for surgical patients because the initiation of enteral feeding tends to be delayed for most surgical patients. Despite multiple guidelines recommending the early use of EN in ICUs,[6, 7] only half of the pre-scribed calories can be delivered to patients.[15] Because of this failure, SPN might play a role

**Table 3. Clinical outcomes.**

| Outcome | Protocol (n = 85) | Control (n = 85) | p-value |
|---|---|---|---|
| Duration of mechanical ventilation (days) | 4.6 (2.7–12.2) | 4.9 (2.4–10.7) | 0.826 |
| New infection, n (%) | | | 0.586 |
| Any infection | 28 (32.9) | 29 (34.1) | |
| Pneumonia | 17 (20.0) | 12 (14.1) | |
| ICU mortality, n (%) | 10 (11.8) | 15 (17.6) | 0.279 |
| Hospital mortality, n (%) | 23 (27.1) | 22 (25.9) | 0.862 |
| ICU LOS (days), median (IQR) | 6.2 (4.3–13.3) | 6.3 (3.6–13.1) | 0.574 |
| Hospital LOS (days), median (IQR) | 27.3 (15.2–44.4) | 27.1 (12.2–44.2) | 0.618 |

Abbreviations: ICU, intensive care unit; LOS, length of stay.

**Table 4. Nutrition-related complications.**

| Outcome | Protocol (n = 85) | Control (n = 85) | P-value |
|---|---|---|---|
| Hyperglycemia, n (%) | 18 (21.2) | 20 (23.5) | 0.713 |
| Hypoglycemia, n (%) | 2 (2.4) | 2 (2.4) | 1.000 |
| Vomiting, n (%) | 3 (3.5) | 3 (3.5) | 1.000 |
| Abdominal distension, n (%) | 2 (2.4) | 4 (4.7) | 0.414 |
| Aspiration, n (%) | 1 (1.2) | 0 | 0.316 |
| GRV > 250 ml, n (%) | 22 (25.9) | 22 (25.9) | 1.000 |

Abbreviations: GRV, gastric residual volume.

in decreasing the calorie debt, especially if patients are suffering from malnutrition.[7, 10, 16] The European Society for Parenteral and Enteral Nutrition guidelines recommend initiating EN within 24–48 hours for patients who are not expected to receive full oral nutrition within 3 days, and initiating SPN if the EN levels are not at goal in 48 hours.[7] In a recent randomized controlled trial[10] designed to deliver SPN only to critically ill patients who were either underweight or overweight, a significantly higher calorie intake was demonstrated for the SPN +EN arm than for the EN-alone arm. Not surprisingly, that study found that the surgical ICU patients received a poorer EN nutrition delivery but had a significantly greater increase in calorie and protein delivery when receiving SPN than did the medical ICU patients. Moreover, a subgroup analysis revealed that the patients with the highest ICU admission nutrition risk (a NUTRIC score ≥ 5) appeared to gain the most benefit from SPN. In comparison, the majority of the patients (70%) in the current study had a BMI < 25 and a NUTRIC score ≥ 5 on admission; such a population should benefit from an SPN protocol. The PN in our study was initiated from the first day after ICU admission, and the commencement times for the protocol and non-protocol groups were not significantly different.

A previous prospective cohort study[17] at our SICU reported the time to start enteral feeding was approximately 4.5 days. In contrast, the present study showed shorter times of 1.9 days for the protocol group and 2.3 days for the control group. Almost half of the patients in both groups were given EN within 48 hours. As expected in this study of a SICU population, the surgical conditions were the major issues delaying both the initiation of the EN feeding and a step-up of the EN. In addition, more than half of the patients in this trial underwent intra-abdominal surgery, which might have precluded early EN feeding. Nevertheless, the time to initiate the EN was improved compared with the results from previous decades. With regard to the received calories per day during the first 7 days after admission, the average daily calorie intake was approximately 750 kcal, and there was no significant difference between the two arms. Still, a previous study[18] from the THAI-SICU database established a much lower level of total daily calories received (228 kcal) than did the current study. An increase in awareness of nutrition therapy and the knowledge gained from intensive course training might be the key factors that have resulted in the improvement in the effectiveness of feeding. Despite that, the effectiveness of the feedings of the protocol and non-protocol groups did not differ significantly, and there were consequently no significant differences in their clinical outcomes (new infections, hospital and ICU LOS, hospital and ICU mortality rates, and nutrition-related complications). Nonetheless, several trials have shown that the proper implementation of enteral-feeding protocols is able to reduce septic morbidity, ICU and hospital LOS, the need for mechanical ventilation, and mortality.[19–22] As those trials were observational in nature, the impact of an EN-feeding protocol requires further evaluation.

The strength of the present study includes its randomized controlled trial design, which employed a pragmatic nutrition protocol based on guideline recommendations[7] and made comparisons with a non-protocol arm (the control group). Moreover, the study established that the administration of early SPN did not result in an increased risk of infection, as had been hypothesized in a previous trial.[23]

Some limitations of the current research need to be addressed. Despite its randomized controlled trial design, the study was conducted at the same SICU for both the protocol and non-protocol groups; this meant that the Hawthorne effect was unavoidable. In addition, the attending physicians were assigned to take care of the patients in both study arms, and they were also the experts in nutrition therapy in the ICU. As a result, their routine nutrition therapy practice might not have differed greatly from the protocol. Finally, although our study did not show any differences in terms of the effectiveness of the feedings of the protocol and non-protocol groups, it was our observation that the clinical practice had changed in that there was an increased awareness of the need to prescribe adequate nutrition to the critically ill patients.

## Conclusions

The implementation of the nutrition protocol in the Surgical ICU did not demonstrate any significant improvement in either the effectiveness of feeding or the clinical outcomes, compared with the usual nutrition management practices.

## Supporting information

**S1 Fig. Flow chart of Siriraj surgical intensive care unit (SICU) nutrition protocol.**
(JPG)

**S2 Fig. Flow chart of Siriraj supplemental parenteral nutrition (SPN) protocol.**
(JPG)

**S3 Fig. Flow chart of Siriraj gastrointestinal (GI) intolerance protocol.**
(JPG)

**S1 Checklist. CONSORT 2010 checklist of information to include when reporting a randomised trial.**
(DOC)

**S1 Data.**
(PDF)

## Acknowledgments

The authors also thank Nichapat Thongkaew, a research assistant, for her invaluable help with the paperwork.

## Author Contributions

**Conceptualization:** Onuma Chaiwat.

**Data curation:** Pornrat Chinda, Pulyamon Poomthong, Puriwat Toadithep.

**Formal analysis:** Chayanan Thanakiattiwibun.

**Investigation:** Pornrat Chinda, Onuma Chaiwat.

**Methodology:** Pornrat Chinda, Onuma Chaiwat.

**Project administration:** Onuma Chaiwat.

**Writing – original draft:** Pornrat Chinda, Pulyamon Poomthong, Puriwat Toadithep, Chayanan Thanakiattiwibun, Onuma Chaiwat.

**Writing – review & editing:** Pornrat Chinda, Pulyamon Poomthong, Puriwat Toadithep, Onuma Chaiwat.

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
