## [Decision Letter · Decision Letter 0]

4 Feb 2020

PONE-D-19-33784

The implementation of a nutrition protocol in a surgical intensive care unit; a randomized controlled trial at a tertiary care hospital

PLOS ONE

Dear Professor Chaiwat,

Thank you for submitting your manuscript to PLOS ONE. After careful consideration, we feel that it has merit but does not fully meet PLOS ONE’s publication criteria as it currently stands. Therefore, we invite you to submit a revised version of the manuscript that addresses the points raised during the review process.

Editir comments,

Although the study is methodologically well developed, some problems make its result difficult to interpret and decrease its relevance. Please, try to clarify some points:

1. Differences in outcomes will not be seen, since there was no difference between the protocol and non-protocol groups. Wouldn't it be interesting to see if the diet receiving quartiles had different outcomes?

2. It was not possible to understand which method of nutritional assessment was used. Was NRS, MUST or Nutric used for the diagnosis of malnutrition? If they gave different diagnoses, which one would be considered?

3. How was the height of the patients obtained (used for the ideal weight formula)?

4. Did you use the ideal weight even in the patient with a BMI below 30 kg / m? The ESPEN guideline, 2019, indicates the use of current weight.

5. what was weight used for the screening / nutritional assessment tools? And how was this weight obtained?

I believe that these questions need to be clarified before your paper can be accepted for publication in a high-impact journal, such as Plos One.

We would appreciate receiving your revised manuscript by 03 rd April 2020. To enhance the reproducibility of your results, we recommend that if applicable you deposit your laboratory protocols in protocols.io, where a protocol can be assigned its own identifier (DOI) such that it can be cited independently in the future. For instructions see: http://journals.plos.org/plosone/s/submission-guidelines#loc-laboratory-protocols

We look forward to receiving your revised manuscript.

Kind regards,

Daniela Ponce, PhD

Academic Editor

PLOS ONE

Journal Requirements:

3. We note you have included a table to which you do not refer in the text of your manuscript. Please ensure that you refer to Table 3 and 4 in your text; if accepted, production will need this reference to link the reader to the Table.

Reviewers' comments:

Reviewer's Responses to Questions

**Comments to the Author**

1. Is the manuscript technically sound, and do the data support the conclusions?

Reviewer #1: Partly

Reviewer #2: Partly

Reviewer #3: Yes

2. Has the statistical analysis been performed appropriately and rigorously? 

Reviewer #1: No

Reviewer #2: Yes

Reviewer #3: Yes

3. Have the authors made all data underlying the findings in their manuscript fully available?

Reviewer #1: Yes

Reviewer #2: Yes

Reviewer #3: Yes

4. Is the manuscript presented in an intelligible fashion and written in standard English?

Reviewer #1: Yes

Reviewer #2: Yes

Reviewer #3: Yes

5. Review Comments to the Author

Reviewer #1: The authors performed a randomized controlled trial to evaluate the efficacy of a surgical ICU nutrition protocol and to compare the groups (protocol and non) according to clinical outcomes, including 170 patients. There are some recommendations that might be useful to consider:

- It is important to verify the use of the words efficacy x effectiveness throughout the text. In abstract it was described that the aim was to evaluate the efficacy of the protocol and to compare outcomes between groups. In the background, the aim was to develop a nutrition protocol and to compare the effectiveness of nutritional therapy and clinical outcomes. It’s necessary to standardize this information.

- Indicate in the aim which clinical outcomes were evaluated (hospital mortality, LOS,...) - line 56

- Indicate the reference used for the predicted body weight - lines 74 and 75

- Indicate the reference of the scores: NRS 2002, MUST and NUTRIC - lines 76, 77 and 78

- Provide more information about the control group. Was any nutritional risk screening applied? Is here any recommendation of the initiation of medical nutrition therapy? Or did the attending physicians decide when and how?

- Provide more information about the follow-up and outcomes (mortality, LOS, ...). How and when were they assessed?

- Describe the abbreviation of BMI (line 140)

- Line 145 - p value is different from that shown in table 1

- Line 146 - protocol group (%) and p value were different from those in table 1

- Line 155 - p value is different from that shown in table 2

- Table 2: describe the abbreviation of NPO

- Cite the tables 3 and 4 in the results

- Table 4: describe the abbreviation of GRV

- Line 187 - PN? According to ESPEN, EN is the first choice for nutrition support during the first three to four days after ICU admission. If EN is not feasible or is insufficient after three days, PN should be prescribed

- CONSORT checklist - item 12b: The authors listed that additional analyses were in page 7. What analyses were done in addition to the others presented?

- CONSORT checklist - item 21: It is important to include the external validity and applicability of the trial findings.

Reviewer #2: The present manuscript shows the results of a clinical trial where a protocol for improving nutrition therapy in surgical ICU patients was tested.

1)While the use of protocols to improve clinical nutrition is still an issue, I presume that this kind of strategy should be useful when none management has been used in the current practice. Therefore, I suggest the authors to state the routine management of ICU surgical patients. If the current routine and the protocol have similar strategies, the clinical trial would not be the best option to state that protocol-based nutrition therapy doesn’t work.

2) Lines 15 and 16 abstract and 152 -153 – suggest the time to commence the nutrition therapy tended to be shorter in the protocol group. However, the P-value is too high to say tendency.

3) 125 - Which outcome do you consider to calculate the sample size?

4) Line 159 – The authors observed that surgical conditions delay enteral nutrition delivery in almost 50% of patients. May you explain better about this? Did the patients develop metabolic ileus, distension, fistula?

5) Do you use Enhanced recovery after surgery strategies, such as reducing volume overload, reducing fasting time, etc.? Regarding pre and intra-surgical management, my question here is: Is the protocol that doesn’t work? Or in both groups, the patients have not been well prepared to receive enteral nutrition in the first 24-48 days.

5) May you explain about diarrhea? There is a protocol involving this complication that could reduce nutrition delivery, but I couldn’t find data about it.

6) Table 2 – page 18. The hypocaloric diet is expected for the first days of ICU hospitalization. However, the amount of protein is too low and different between the goups. May you explain about it?

7) Which type of vascular surgery? Large surgeries for an aortic replacement, for example, are completely different from varicose vein. While vascular surgery was more common in the protocol group, could the type of surgery have any influence?

Reviewer #3: The present study aimed to evaluate the effectiveness of a surgical nutrition protocol in the ICU and to compare hospital mortality, length of hospital stay and length of stay in the ICU of the protocol and non-protocol groups. Although the study is methodologically well developed, some problems make its result difficult to interpret.

1- how to explain the ineffectiveness of the nutritional protocol? Both groups (protocol and non-protocol) had around 50% of patients receiving nutritional therapy in the first 48 hours. This rate is low, but is close to that reported in recent literature (PLoS One. 2017 Aug 3; 12 (8): e0182393. Doi: 10.1371 / journal.pone.0182393; World J Crit Care Med. 2017 Feb 4; 6 ( 1): 56-64.doi: 10.5492 / wjccm.v6.i1.56; Asia Pac J Clin Nutr. 2017 Jan; 26 (1): 27-35.doi: 10.6133 / apjcn.122015.01; among others). So, the central question is: why don't we feed almost 50% of patients?

2- Differences in outcomes will not be seen, since there was no difference between the protocol and non-protocol groups. Wouldn't it be interesting to see if the diet receiving quartiles had different outcomes?

Minor considerations:

1 - it was not possible to understand which method of nutritional assessment was used. Was NRS, MUST or Nutric used for the diagnosis of malnutrition? If they gave different diagnoses, which one would be considered?

2- how was the height of the patients obtained (used for the ideal weight formula)?

3- Did you use the ideal weight even in the patient with a BMI below 30 kg / m? The ESPEN guideline, 2019, indicates the use of current weight.

4- What weight was used for the screening / nutritional assessment tools? And how was this weight obtained?

6. PLOS authors have the option to publish the peer review history of their article (what does this mean?). If published, this will include your full peer review and any attached files.

Reviewer #1: No

Reviewer #2: No

Reviewer #3: No

---

## [Author Response · Author response to Decision Letter 0]

26 Feb 2020

Editor comments:

Although the study is methodologically well developed, some problems make its result difficult to interpret and decrease its relevance. Please, try to clarify some points:

1. Differences in outcomes will not be seen, since there was no difference between the protocol and non-protocol groups. Wouldn't it be interesting to see if the diet receiving quartiles had different outcomes?

Answer: When we considered 50%, 40% and 30% calorie target received on Day 4 instead of 60% calorie target received on Day 4 as we had reported on table 2, the difference between protocol and control groups regarding calorie target received on Day 4 remains non-significant. 

Nutrition outcomes Protocols (n=85) Control (n=85) p-value

50% calorie target received on Day 4, n (%) 42(49.4%) 29(34.1%) 0.062

40% calorie target received on Day 4, n (%) 47(55.3%) 36(42.4%) 0.125

30% calorie target received on Day 4, n (%) 54(63.5%) 46(54.1%) 0.280

2. It was not possible to understand which method of nutritional assessment was used. Was NRS, MUST or Nutric used for the diagnosis of malnutrition? If they gave different diagnoses, which one would be considered?

Answer: Since no specific ICU nutritional score has been validated thus far. The existing nutritional screening tools NRS 2002 and the malnutrition universal screening tool (MUST) score have not been designed specifically for critically ill patients, we decided to present both NRS and MUST for baseline nutrition assessment at ICU admission and there were not significantly different between protocol and non-protocol groups. However, we used Nutric at day 4 to define malnutrition in order to commence parenteral nutrition.

3. How was the height of the patients obtained (used for the ideal weight formula)?

Answer: We measured the height by the tape measure in patients who did not have the height recorded in the medical records or used the height that recorded in the medical record in patients who had that records.

4. Did you use the ideal weight even in the patient with a BMI below 30 kg / m? The ESPEN guideline, 2019, indicates the use of current weight.

Answer: Yes, we used the ideal body weight in all patients. We did this study before the recommendation from ESPEN 2019. However, in the majority of Thai population the actual and ideal body weight is not much different and both the body weight and BMI were not significantly different between groups.

5. What was weight used for the screening / nutritional assessment tools? And how was this weight obtained?

Answer: In the case of males, the formula used to calculate their predicted body weight (in kg) was (50.0 + 0.91*[height in cm – 152.4]). As to females, the predicted body weight (in kg) was given by (45.5 + 0.91*[height in cm – 152.4]) (line 73-75). The height was obtained as mentioned in question 3.

Journal Requirements:

Answer: Thank you for your suggestion. We did it as suggestion.

Answer: Thank you for your suggestion. We send to language editing by a native English speaker and edited manuscript (Mr. David William Park, faculty of Medicine Siriraj hospital) formatting according to the guidelines of journal.

3. We note you have included a table to which you do not refer in the text of your manuscript. Please ensure that you refer to Table 3 and 4 in your text; if accepted, production will need this reference to link the reader to the Table.

Answer: We added as suggestion. We referred to Table 3 and 4 in manuscript (page 9, results, paragraph 3, line 168 and line 172).

Reviewers' comments:

Reviewer's Responses to Questions

Comments to the Author

1. Is the manuscripts technically sound, and do the data support the conclusions?

Reviewer #1: Partly

Reviewer #2: Partly

Reviewer #3: Yes

2. Has the statistical analysis been performed appropriately and rigorously?

Reviewer #1: No

Reviewer #2: Yes

Reviewer #3: Yes

3. Have the authors made all data underlying the findings in their manuscript fully available?

Reviewer #1: Yes

Reviewer #2: Yes

Reviewer #3: Yes

4. Is the manuscript presented in an intelligible fashion and written in standard English?

Reviewer #1: Yes

Reviewer #2: Yes

Reviewer #3: Yes

5. Review Comments to the Author

Reviewer #1: The authors performed a randomized controlled trial to evaluate the efficacy of a surgical ICU nutrition protocol and to compare the groups (protocol and non) according to clinical outcomes, including 170 patients. There are some recommendations that might be useful to consider:

- It is important to verify the use of the words efficacy x effectiveness throughout the text. In abstract it was described that the aim was to evaluate the efficacy of the protocol and to compare outcomes between groups. In the background, the aim was to develop a nutrition protocol and to compare the effectiveness of nutritional therapy and clinical outcomes. It’s necessary to standardize this information.

Answer: Thank you for your suggestion. We changed the “efficacy to effectiveness” in entire manuscript.

- Indicate in the aim which clinical outcomes were evaluated (hospital mortality, LOS,...) - line 56

Answer: We added as suggestion. (nutritional related complications, hospital and ICU mortality and length of stay).

- Indicate the reference used for the predicted body weight - lines 74 and 75

Answer: We added as suggestion (page 5, materials and methods, line 75).

Reference:

11. Brower RG, Matthay MA, Morris A, Schoenfeld D, Thompson BT, Wheeler A. Ventilation with lower tidal volumes as compared with traditional tidal volumes for acute lung injury and the acute respiratory distress syndrome. The New England journal of medicine. 2000;342(18):1301-8.

- Indicate the reference of the scores: NRS 2002, MUST and NUTRIC - lines 76, 77 and 78

Answer: We added as suggestion (page 5, materials and methods, line 76, 77, 78).

Reference:

12. Kondrup J, Allison SP, Elia M, Vellas B, Plauth M. ESPEN guidelines for nutrition screening 2002. Clin Nutr. 2003;22(4):415-21.

13. Elia M. The 'MUST' report. Nutritional screening for adults: a multidisciplinary responsibility. Development and use of the 'Malnutrition Universal Screening Tool' (MUST) for adults.: British Association for Parenteral and Enteral Nutrition (BAPEN); 2003.

14. Heyland DK, Dhaliwal R, Jiang X, Day AG. Identifying critically ill patients who benefit the most from nutrition therapy: the development and initial validation of a novel risk assessment tool. Critical care (London, England). 2011;15(6):R268.

- Provide more information about the control group. Was any nutritional risk screening applied? Is here any recommendation of the initiation of medical nutrition therapy? Or did the attending physicians decide when and how?

Answer: Nutritional risk screening was not applied routinely for the control group, we calculated for this study without informing the attending staff. No any nutritional guidelines had been available in SICU before, the nutritional therapy was made by the attending physicians at any time or any methods.

- Provide more information about the follow-up and outcomes (mortality, LOS, ...). How and when were they assessed?

Answer: ICU, In-hospital mortality, LOS and nutrition related complication, data was retrieved from the researcher (prospective data mining from medical records) in the study (PC, PP, PT)

- Describe the abbreviation of BMI (line 140)

Answer: We did as suggestion.

- Line 145 - p value is different from that shown in table 1

Answer: Thank you for pointing out this error. We have checked the results as suggested (page 8, results, paragraph 1, line 145).

- Line 146 - protocol group (%) and p value were different from those in table 1

Answer: Thank you for pointing out this error. We have checked the results as suggested (page 8, results, paragraph 1, line 146).

- Line 155 - p value is different from that shown in table 2

Answer: Thank you for pointing out this error. We have checked the results as suggested (page 8, results, paragraph 2, line 154-162).

- Table 2: describe the abbreviation of NPO

Answer: We did as suggestion.

- Cite the tables 3 and 4 in the results

Answer: We did as suggestion.

- Table 4: describe the abbreviation of GRV

Answer: We did as suggestion.

- Line 187 - PN? According to ESPEN, EN is the first choice for nutrition support during the first three to four days after ICU admission. If EN is not feasible or is insufficient after three days, PN should be prescribed

Answer: Thank you for the correction, we have changed PN to EN.

- CONSORT checklist - item 12b: The authors listed that additional analyses were in page 7. What analyses were done in addition to the others presented?

Answer: Thank you for pointing out. We have corrected the CONSORT checklist item 12b…No additional analyses.

- CONSORT checklist - item 21: It is important to include the external validity and applicability of the trial findings.

Answer: We pointed out in the discussion, line 232-235.

Reviewer #2: The present manuscript shows the results of a clinical trial where a protocol for improving nutrition therapy in surgical ICU patients was tested.

1) While the use of protocols to improve clinical nutrition is still an issue, I presume that this kind of strategy should be useful when none management has been used in the current practice. Therefore, I suggest the authors to state the routine management of ICU surgical patients. If the current routine and the protocol have similar strategies, the clinical trial would not be the best option to state that protocol-based nutrition therapy doesn’t work.

Answer: We totally agree. Basically, nutritional risk screening was not applied routinely for the control group, we calculated the scores for this study without informing the attending staff. No any nutritional guidelines had been available in SICU before, the nutritional therapy was made by the attending physicians at any time or any methods. However, the faculty provided education regarding the nutrition therapy to critical care fellow and residents that occurred during the study period. Although we haven’t had any protocol before, the healthcare providers might have better knowledge regarding nutrition therapy. 

2) Lines 15 and 16 abstract and 152 -153 – suggest the time to commence the nutrition therapy tended to be shorter in the protocol group. However, the P-value is too high to say tendency.

Answer: We corrected from tended to “was”.

3) 125 - Which outcome do you consider to calculate the sample size?

Answer: It was estimated that 50% of the patients in the control group and 75% in the protocol group would achieve 100% of the target calories with statistical significance. We used the effectiveness of nutrition therapy in term of achieve 100% of the target calories.

4) Line 159 – The authors observed that surgical conditions delay enteral nutrition delivery in almost 50% of patients. May you explain better about this? Did the patients develop metabolic ileus, distension, fistula?

Answer: We added more detail about surgical conditions. Moreover, the main reason for interrupting or stepping up the EN was the patients’ surgical conditions including gastroparesis, ileus and surgical complication (leakage and sepsis), followed by hemodynamic instability. (page 9, paragraph 2, line 161)

5) Do you use Enhanced recovery after surgery strategies, such as reducing volume overload, reducing fasting time, etc.? Regarding pre and intra-surgical management, my question here is: Is the protocol that doesn’t work? Or in both groups, the patients have not been well prepared to receive enteral nutrition in the first 24-48 days.

Answer: During the study period, our hospital did not use the ERAS protocol. We just implemented the ERAS on November, 2019. We tried our best to commence EN as soon as possible according to the protocol, however, the decision was depended on each surgeon even we had informed about the protocol. 

5) May you explain about diarrhea? There is a protocol involving this complication that could reduce nutrition delivery, but I couldn’t find data about it.

Answer: Since there was less than 20% of patients in both groups received only EN (table 2), no clinical significant diarrhea (clinical significant stools, diagram 3) was reported from our cohort.

6) Table 2 – page 18. The hypocaloric diet is expected for the first days of ICU hospitalization. However, the amount of protein is too low and different between the groups. May you explain about it?

Answer: We did not know the exact reasons. According to the protocol, we did not target the amount of delivered protein and the mean calories received was quite low due to surgical conditions that preclude early feeding including gastroparesis, ileus and surgical complications (leakage and sepsis). Moreover, the effectiveness of the therapy was reported in terms of the time to commence it, the percentage of patients who received EN within the first 48 hours, the total daily calories received, and the proportion of patients receiving > 60% of the target calories on Day 4 of their ICU admission.

7) Which type of vascular surgery? Large surgeries for an aortic replacement, for example, are completely different from varicose vein. While vascular surgery was more common in the protocol group, could the type of surgery have any influence?

Answer: The majority of vascular surgery in our SICU was major aortic surgery including infrarenal aortic aneurysm surgery, the acute limb ischemia from arterial occlusion and the bypass surgery. Although the number of vascular surgeries was higher in protocol group, the severity of patients measured by APACHE II and SOFA score was not significantly different between groups.

Reviewer #3: The present study aimed to evaluate the effectiveness of a surgical nutrition protocol in the ICU and to compare hospital mortality, length of hospital stay and length of stay in the ICU of the protocol and non-protocol groups. Although the study is methodologically well developed, some problems make its result difficult to interpret.

1- how to explain the ineffectiveness of the nutritional protocol? Both groups (protocol and non-protocol) had around 50% of patients receiving nutritional therapy in the first 48 hours. This rate is low, but is close to that reported in recent literature (PLoS One. 2017 Aug 3; 12 (8): e0182393. Doi: 10.1371 / journal.pone.0182393; World J Crit Care Med. 2017 Feb 4; 6 ( 1): 56-64.doi: 10.5492 / wjccm.v6.i1.56; Asia Pac J Clin Nutr. 2017 Jan; 26 (1): 27-35.doi: 10.6133 / apjcn.122015.01; among others). So, the central question is: why don't we feed almost 50% of patients?

Answer: The main reason for interrupting or stepping up the EN was the patients’ surgical conditions including gastroparesis, ileus and surgical complication (leakage and sepsis), followed by hemodynamic instability. This was the main reason for ineffectiveness of a surgical nutrition protocol in SICU. In addition, the decision to start feeding partly depended on surgeon that we mentioned in the guidelines (with surgeon’s permission, table 1).

2- Differences in outcomes will not be seen, since there was no difference between the protocol and non-protocol groups. Wouldn't it be interesting to see if the diet receiving quartiles had different outcomes?

Answer: When we considered 50%, 40% and 30% calorie target received on Day 4 instead of 60% calorie target received on Day 4 as we had reported on table 2, the differences between protocol and control group regarding calorie target received on Day 4 remains non-significant.

Nutrition outcomes Protocols (n=85) Control (n=85) p-value

50% calorie target received on Day 4, n (%) 42(49.4%) 29(34.1%) 0.062

40% calorie target received on Day 4, n (%) 47(55.3%) 36(42.4%) 0.125

30% calorie target received on Day 4, n (%) 54(63.5%) 46(54.1%) 0.280

Minor considerations:

1 - it was not possible to understand which method of nutritional assessment was used. Was NRS, MUST or Nutric used for the diagnosis of malnutrition? If they gave different diagnoses, which one would be considered?

Answer: Since no specific ICU nutritional score has been validated thus far. The existing nutritional screening tools NRS 2002 and the malnutrition universal screening tool (MUST) score have not been designed specifically for critically ill patients, we decided to present both NRS and MUST for baseline nutrition assessment at ICU admission and there were not significantly different between protocol and non-protocol groups. However, we used Nutric at day 4 to define malnutrition in order to commence parenteral nutrition.

2- how was the height of the patients obtained (used for the ideal weight formula)?

Answer: We measured the height by the tape measure in patients who did not have the height recorded in the medical record or used the height that recorded in the medical record in patients who had that record.

3- Did you use the ideal weight even in the patient with a BMI below 30 kg / m? The ESPEN guideline, 2019, indicates the use of current weight.

Answer: Yes, we used an ideal body weight in all patients. We did this study before the recommendation from ESPEN 2019. However, in the majority of Thai population the actual and ideal body weight is not much different and both the body weight and BMI were not significantly different between groups.

4- What weight was used for the screening / nutritional assessment tools? And how was this weight obtained?

Answer: In the case of males, the formula used to calculate their predicted body weight (in kg) was (50.0 + 0.91*[height in cm – 152.4]). As to females, the predicted body weight (in kg) was given by (45.5 + 0.91*[height in cm – 152.4]) (line 73-75). The height was obtained as mentioned in question 2.

6. PLOS authors have the option to publish the peer review history of their article (what does this mean?). If published, this will include your full peer review and any attached files.

Do you want your identity to be public for this peer review? For information about this choice, including consent withdrawal, please see our Privacy Policy.

Reviewer #1: No

Reviewer #2: No

Reviewer #3: No

---

## [Decision Letter · Decision Letter 1]

1 Apr 2020

The implementation of a nutrition protocol in a surgical intensive care unit; a randomized controlled trial at a tertiary care hospital

PONE-D-19-33784R1

Dear Professor,

We are pleased to inform you that your manuscript has been judged scientifically suitable for publication and will be formally accepted for publication once it complies with all outstanding technical requirements.

With kind regards,

Daniela Ponce, PhD

Academic Editor

PLOS ONE

Additional Editor Comments (optional):

Reviewers' comments:

Reviewer's Responses to Questions

**Comments to the Author**

1. If the authors have adequately addressed your comments raised in a previous round of review and you feel that this manuscript is now acceptable for publication, you may indicate that here to bypass the “Comments to the Author” section, enter your conflict of interest statement in the “Confidential to Editor” section, and submit your "Accept" recommendation.

Reviewer #1: All comments have been addressed

Reviewer #2: All comments have been addressed

2. Is the manuscript technically sound, and do the data support the conclusions?

Reviewer #1: Yes

Reviewer #2: Yes

3. Has the statistical analysis been performed appropriately and rigorously? 

Reviewer #1: Yes

Reviewer #2: Yes

4. Have the authors made all data underlying the findings in their manuscript fully available?

Reviewer #1: Yes

Reviewer #2: Yes

5. Is the manuscript presented in an intelligible fashion and written in standard English?

Reviewer #1: Yes

Reviewer #2: Yes

6. Review Comments to the Author

Reviewer #1: The authors accepted the reviewers suggestions and made adjustments. Therefore, some points could still be elucidated in the text to make it clear:

- Information about the outcomes (Data was retrieved from the researcher on medical records) - line 89

- Information about the control group (Nutritional risk screening was not applied routinely for the control group. No any nutritional guidelines had been available in SICU before, the nutritional therapy was made by the attending physicians at any time or any methods) - line 99

Reviewer #2: It is an interesting and well-conducted trial.

I still concern about the objectives of a protocol implementation. To the top of my head, nutritional protocols aim to deliver adequate energy and protein as soon as possible. In this trial, the protocol did not accomplish the objective. Both groups received the same energy amount, and the protocol group received less protein.

However, the authors discussed about this in the limmitation of the study.

7. PLOS authors have the option to publish the peer review history of their article (what does this mean?). If published, this will include your full peer review and any attached files.

Reviewer #1: No

Reviewer #2: No

---

## [Editor Report · Acceptance letter]

3 Apr 2020

PONE-D-19-33784R1 

The implementation of a nutrition protocol in a surgical intensive care unit; a randomized controlled trial at a tertiary care hospital 

Dear Dr. Chaiwat:

I am pleased to inform you that your manuscript has been deemed suitable for publication in PLOS ONE. Congratulations! Your manuscript is now with our production department. 

With kind regards,

on behalf of

Dr. Daniela Ponce 

Academic Editor

PLOS ONE